# Fast and Simple Spectral Clustering in Theory and Practice

**Peter Macgregor**
School of Informatics
University of Edinburgh
`peter.macgregor@ed.ac.uk`

## Abstract

Spectral clustering is a popular and effective algorithm designed to find $k$ clusters in a graph $G$. In the classical spectral clustering algorithm, the vertices of $G$ are embedded into $\mathbb{R}^k$ using $k$ eigenvectors of the graph Laplacian matrix. However, computing this embedding is computationally expensive and dominates the running time of the algorithm. In this paper, we present a simple spectral clustering algorithm based on a vertex embedding with $O(\log(k))$ vectors computed by the power method. The vertex embedding is computed in nearly-linear time with respect to the size of the graph, and the algorithm provably recovers the ground truth clusters under natural assumptions on the input graph. We evaluate the new algorithm on several synthetic and real-world datasets, finding that it is significantly faster than alternative clustering algorithms, while producing results with approximately the same clustering accuracy.

## 1 Introduction

Graph clustering is an important problem with numerous applications in machine learning, data science, and theoretical computer science. Spectral clustering is a popular graph clustering algorithm with strong theoretical guarantees and excellent empirical performance. Given a graph $G$ with $n$ vertices and $k$ clusters, the classical spectral clustering algorithm consists of the following two high-level steps [26, 33].

1. Embed the vertices of $G$ into $\mathbb{R}^k$ according to $k$ eigenvectors of the graph Laplacian matrix.
2. Apply a $k$-means clustering algorithm to partition the vertices into $k$ clusters.

Recent work shows that if the graph has a well-defined cluster structure, then the clusters are well-separated in the spectral embedding and the $k$-means algorithm will return clusters which are close to optimal [21, 28].

The main downside of this algorithm is the high computational cost of computing $k$ eigenvectors of the graph Laplacian matrix. In this paper, we address this computational bottleneck and propose a new fast spectral clustering algorithm which avoids the need to compute eigenvectors while maintaining excellent theoretical guarantees. Moreover, our proposed algorithm is simple, fast, and effective in practice.

### 1.1 Sketch of Our Approach

In this section, we introduce the high-level idea of this paper, which is also illustrated in Figure 1. We begin by considering a recent result of Makarychev et al. [23] who show that a random projection of data into $O(\log(k))$ dimensions preserves the $k$-means objective function for all partitions of the

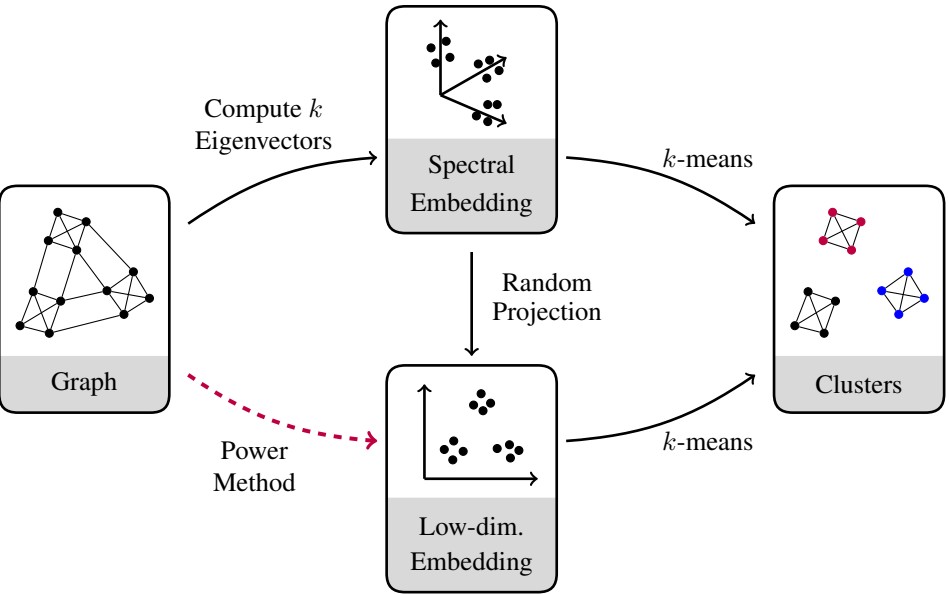

Figure 1: An illustration of the steps of the spectral clustering algorithm, and the contribution of this paper. We are given a graph as input. In classical spectral clustering we follow the top path: we compute the spectral embedding and apply a $k$-means algorithm to find clusters. Through the recent result of Makarychev et al. [23], we can project the embedded points into $O(\log(k))$ dimensions and obtain approximately the same clustering. In this paper, we show that it is possible to compute the low-dimensional embedding directly with the power method, skipping the computationally expensive step of computing $k$ eigenvectors.

data, with high probability. Since the final step of spectral clustering is to apply $k$-means, we might consider the following alternative spectral clustering algorithm which will produce roughly the same output as the classical algorithm.

1. Embed the vertices of $G$ into $\mathbb{R}^k$ according to $k$ eigenvectors of the graph Laplacian matrix.

2. Randomly project the embedded points into $O(\log(k))$ dimensions.

3. Apply a $k$-means clustering algorithm to partition the vertices into $k$ clusters.

Of course, this does not avoid the expensive eigenvector computation and so it is not immediately clear that this random projection can be used to improve the spectral clustering algorithm.

The key technical element of our paper is a proof that we can efficiently approximate a random projection of the spectral embedding without computing the spectral embedding itself. For this, we use the power method, which is a well-known technique in numerical linear algebra for approximating the dominant eigenvalue of a matrix [11]. We propose the following simple algorithm (formally described in Algorithm 2).

1. Embed the vertices of $G$ into $O(\log(k))$ dimensions using $O(\log(k))$ random vectors computed with the power method.

2. Apply a $k$-means clustering algorithm to partition the vertices into $k$ clusters.

We prove that the projection obtained using the power method is approximately equivalent to a random projection of the spectral embedding. Then, by carefully applying the techniques developed by Makarychev et al. [23] and Macgregor and Sun [21], we obtain a theoretical bound on the number of vertices misclassified by our proposed algorithm. Moreover, the time complexity of step 1 is nearly linear in the size of the graph, and the algorithm is fast in practice. The formal theoretical guarantee is given in Theorem 3.1.

## 1.2 Related Work

This paper is closely related to a sequence of recent results which prove an upper bound on the number of vertices misclassified by the classical spectral clustering algorithm [15, 21, 24, 28]. While we will directly compare our result with these in a later section, our proposed algorithm has a much faster running time than the classical spectral clustering algorithm, and has similar theoretical guarantees.

Boutsidis et al. [5] also study spectral clustering using the power method. Our result improves on theirs in two respects. Firstly, our algorithm is faster since we compute $O(\log(k))$ vectors rather than $k$ vectors and their algorithm includes an additional singular value decomposition step. Secondly, we give a theoretical upper bound on the total number of vertices misclassified by our algorithm.

Makarychev et al. [23] generalise the well-known Johnson-Lindenstrauss lemma [13] to show that random projections of data into $O(\log(k))$ dimensions preserves the $k$-means objective, and we make use of their result in our analysis.

Macgregor and Sun [21] show that for graphs with certain structures of clusters, spectral clustering with fewer than $k$ eigenvectors performs better than using $k$ eigenvectors. In this paper, we present the first proof that embedding with $O(\log(k))$ vectors is sufficient to find $k$ clusters with spectral clustering.

Other proposed methods for fast spectral clustering include the Nystrom method [6] and using a 'pre-clustering' step to reduce the number of data points to be clustered [35]. These methods lack rigorous theoretical guarantees on the accuracy of the returned clustering. Moreover, our proposed algorithm is significantly simpler to implement that the alternative methods.

## 2 Preliminaries

Let $G = (V, E, w)$ be a graph with $n = |V|$ and $m = |E|$. For any $v \in V$, the degree of $v$ is given by $d(v) = \sum_{u \neq v} w(u, v)$. For any $S \subset V$, the volume of $S$ is given by $\mathrm{vol}(S) = \sum_{u \in S} d(u)$. The Laplacian matrix of $G$ is $\mathbf{L} = \mathbf{D} - \mathbf{A}$ where $\mathbf{D}$ is the diagonal matrix with $\mathbf{D}(i, i) = d(i)$ and $\mathbf{A}$ is the adjacency matrix of $G$. The normalised Laplacian is given by $\mathbf{N} = \mathbf{D}^{-\frac{1}{2}} \mathbf{L} \mathbf{D}^{-\frac{1}{2}}$. We always use $\lambda_1 \leq \lambda_2 \leq \ldots \leq \lambda_n$ to be the eigenvalues of $\mathbf{N}$ and the corresponding eigenvectors are $\boldsymbol{f}_1, \ldots, \boldsymbol{f}_n$. For any graph, it holds that $\lambda_1 = 0$ and $\lambda_n \leq 2$ [7]. We will also use the signless Laplacian matrix[1] $\mathbf{M} = \mathbf{I} - (1/2)\mathbf{N}$ and will let $\gamma_1 \geq \ldots \geq \gamma_n$ be the eigenvalues of $\mathbf{M}$. Notice that by the definition of $\mathbf{M}$, we have $\gamma_i = 1 - (1/2)\lambda_i$ and the eigenvectors of $\mathbf{M}$ are also $\boldsymbol{f}_1, \ldots, \boldsymbol{f}_n$. For an integer $k$, we let $[k] = \{1, \ldots k\}$ be the set of all positive integers less than or equal to $k$. Given two sets $A$ and $B$, their symmetric difference is given by $A \triangle B = (A \setminus B) \cup (B \setminus A)$. We call $\{S_i\}_{i=1}^{k}$ a $k$-way partition of $V$ if $S_i \cap S_j = \emptyset$ for $i \neq j$ and $\bigcup_{i=1}^{k} S_i = V$.

Throughout the paper, we use big-O notation to hide constants. For example, we use $l = O(n)$ to mean that there exists a universal constant $c$ such that $l \leq cn$. We sometimes use $\widetilde{O}(n)$ in place of $O(n \log^c(n))$ for some constant $c$. Following [21], we say that a partition $\{S_i\}_{i=1}^{k}$ of $V$ is almost-balanced if $\mathrm{vol}(S_i) = \Theta(\mathrm{vol}(V)/k)$ for all $i \in [k]$.

### 2.1 Conductance and the Graph Spectrum

Given a graph $G = (V, E)$, and a cluster $S \subset V$, the conductance of $S$ is given by

$$\Phi(S) \triangleq \frac{w(S, \overline{S})}{\min\{\mathrm{vol}(S), \mathrm{vol}(\overline{S})\}}$$

where $w(S, \overline{S}) = \sum_{u \in S} \sum_{v \in \overline{S}} w(u, v)$. Then, the $k$-way expansion of $G$ is defined to be

$$\rho(k) \triangleq \min_{\text{partition } C_1, \ldots C_k} \max_i \Phi(C_i).$$

Notice that $\rho(k)$ is small if and only if $G$ can be partitioned into $k$ clusters of low conductance. There is a close connection between the $k$-way expansion of $G$ and the eigenvalues of the graph Laplacian matrix, as shown in the following higher-order Cheeger inequality.

---

[1]The signless Laplacian is usually defined to be $2 \cdot \mathbf{I} - \mathbf{N}$. We divide this by 2 so that the eigenvalues of $\mathbf{M}$ lie between 0 and 1.

**Lemma 2.1** (Higher-Order Cheeger Inequality, [19]). *For a graph G, let $\lambda_1 \leq \ldots \leq \lambda_n$ be the eigenvalues of the normalised Laplacian matrix. Then, for any $k$,*

$$\frac{\lambda_k}{2} \leq \rho(k) \leq O(k^3) \sqrt{\lambda_k}.$$

From this lemma, we can see that an upper bound on $\rho(k)$ and a lower bound on $\lambda_{k+1}$ are sufficient conditions to guarantee that $G$ can be partitioned into $k$ clusters of low conductance, and cannot be partitioned into $k + 1$ clusters. This condition is commonly used in the analysis of graph clustering [15, 21, 24, 28].

## 2.2 The $k$-means Objective

For any matrix $\mathbf{B} \in \mathbb{R}^{n \times d}$, the $k$-means cost of a $k$-way partition $\{A_i\}_{i=1}^k$ of the data points is given by

$$\text{COST}_{\mathbf{B}}(A_1, \ldots, A_k) \triangleq \sum_{i=1}^k \sum_{u \in A_i} \|\mathbf{B}(u,:) - \mu_i\|_2^2,$$

where $\mathbf{B}(u,:)$ is the $u$th row of $\mathbf{B}$ and $\mu_i = (1/|A_i|) \sum_{u \in A_i} \mathbf{B}(u,:)$ is the mean of the points in $A_i$. Although optimising the $k$-means objective is NP-hard, there is a polynomial-time constant factor approximation algorithm [14], and an approximation scheme which is polynomial in $n$ and $d$ with an exponential dependency on $k$ [8, 16]. Lloyds algorithm [20], with $k$-means++ initialisation [3] is an $O(\log(n))$-approximation algorithm which is fast and effective in practice.

## 2.3 The Power Method

The power method is an algorithm which is most often used to approximate the dominant eigenvalue of a matrix [11, 25]. Given some matrix $\mathbf{M} \in \mathbb{R}^{n \times n}$, a vector $\boldsymbol{x}_0 \in \mathbb{R}^n$, and a positive integer $t$, the power method computes the value of $\mathbf{M}^t \boldsymbol{x}_0$ by repeated multiplication by $\mathbf{M}$. The formal algorithm is given in Algorithm 1.

---

**Algorithm 1:** POWERMETHOD($\mathbf{M} \in \mathbb{R}^{n \times n}, \boldsymbol{x}_0 \in \mathbb{R}^n, t \in \mathbb{Z}_{\geq 0}$)

1 **for** $i \in \{1, \ldots, t\}$ **do**
2 $\quad$ $\boldsymbol{x}_i = \mathbf{M} \boldsymbol{x}_{i-1}$
3 **end**
4 **return** $\boldsymbol{x}_t$

---

# 3 Algorithm Description and Analysis

In this section, we present our newly proposed algorithm and sketch the proof of our main result. Omitted proofs can be found in the Appendix. We first prove that if $\mathbf{M}$ has $k$ eigenvalues $\gamma_1, \ldots, \gamma_k$ close to 1, then the power method can be used to compute a random vector in the span of the eigenvectors corresponding to $\gamma_1, \ldots, \gamma_k$.

We then apply this result to the signless Laplacian matrix of a graph to develop a fast spectral clustering algorithm and we bound the number of misclassified vertices when the algorithm is applied to a well-clustered graph.

## 3.1 Approximating a Random Vector with the Power Method

Suppose we are given some matrix $\mathbf{M} \in \mathbb{R}^{n \times n}$ with eigenvalues $1 \geq \gamma_1 \geq \ldots \geq \gamma_n \geq 0$ and corresponding eigenvectors $\boldsymbol{f}_1, \ldots, \boldsymbol{f}_n$. Typically, the power method is used to approximate the dominant eigenvector, $\boldsymbol{f}_1$. In this section, we show that when $\gamma_k$ is sufficiently close to 1, the power method can be used to compute a random vector in the space spanned by $\boldsymbol{f}_1, \ldots, \boldsymbol{f}_k$.

Let $\boldsymbol{x}_0 \in \mathbb{R}^n$ be a random vector chosen according to the $n$-dimensional Gaussian distribution $\mathrm{N}(\mathbf{0}, \mathbf{I})$. We can write $\boldsymbol{x}_0$ as a linear combination of the eigenvectors:

$$\boldsymbol{x}_0 = \sum_{i=1}^{n} a_i \boldsymbol{f}_i,$$

where $a_i = \langle \boldsymbol{x}_0, \boldsymbol{f}_i \rangle$. Then, the vector $\boldsymbol{x}_t$ computed by POWERMETHOD$(\mathbf{M}, \boldsymbol{x}_0, t)$ can be written as

$$\boldsymbol{x}_t = \sum_{i=1}^{n} a_i \gamma_i^t \boldsymbol{f}_i.$$

Informally, if $\gamma_k \geq 1 - c_1$ and $\gamma_{k+1} \leq 1 - c_2$ for sufficiently small $c_1$ and sufficiently large $c_2$, then for a carefully chosen value of $t$ we have $\gamma_i^t \approx 1$ for $i \leq k$ and $\gamma_i^t \approx 0$ for $i \geq k + 1$. This implies that

$$\boldsymbol{x}_t \approx \sum_{i=1}^{k} a_i \boldsymbol{f}_i = \left( \sum_{i=1}^{k} \boldsymbol{f}_i \boldsymbol{f}_i^\mathsf{T} \right) \boldsymbol{x}_0,$$

and we observe that $\left( \sum_{i=1}^{k} \boldsymbol{f}_i \boldsymbol{f}_i^\mathsf{T} \right) \boldsymbol{x}_0$ is a random vector distributed according to a $k$-dimensional Gaussian distribution in the space spanned by $\boldsymbol{f}_1, \ldots, \boldsymbol{f}_k$. We make this intuition formal in Lemma 3.1 and specify the required conditions on $\gamma_k$, $\gamma_{k+1}$ and $t$.

**Lemma 3.1.** *Let* $\mathbf{M} \in \mathbb{R}^{n \times n}$ *be a matrix with eigenvalues* $1 \geq \gamma_1 \geq \ldots \geq \gamma_n \geq 0$ *and corresponding eigenvectors* $\boldsymbol{f}_1, \ldots \boldsymbol{f}_n$. *Let* $\boldsymbol{x}_0 \in \mathbb{R}^n$ *be drawn from the $n$-dimensional Gaussian distribution* $N(\mathbf{0}, \mathbf{I})$. *Let* $\boldsymbol{x}_t = $ POWERMETHOD$(\mathbf{M}, \boldsymbol{x}_0, t)$ *for* $t = \Theta\big(\log(n/\epsilon^2 k)\big)$. *If* $\gamma_k \geq 1 - O\big(\epsilon \cdot \log(n/\epsilon^2 k)^{-1}\big)$ *and* $\gamma_{k+1} \leq 1 - \Omega(1)$, *then with probability at least* $1 - 1/(10k)$,

$$\|\boldsymbol{x}_t - \mathbf{P}\boldsymbol{x}_0\|_2 \leq \epsilon\sqrt{k},$$

*where* $\mathbf{P} = \sum_{i=1}^{k} \boldsymbol{f}_i \boldsymbol{f}_i^\mathsf{T}$ *is the projection onto the space spanned by the first $k$ eigenvectors of* $\mathbf{M}$.

### 3.2 The Fast Spectral Clustering Algorithm

We now introduce the fast spectral clustering algorithm. The algorithm follows the pattern of the classical spectral clustering algorithm, with an important difference: rather than embedding the vertices according to $k$ eigenvectors of the graph Laplacian, we embed the vertices with $\Theta\big(\log(k) \cdot \epsilon^{-2}\big)$ random vectors computed with the power method for the signless graph Laplacian $\mathbf{M}$. Algorithm 2 formally specifies the algorithm, and the theoretical guarantees are given in Theorem 3.1.

**Theorem 3.1.** *Let $G$ be a graph with* $\lambda_{k+1} = \Omega(1)$ *and* $\rho(k) = O\big(\epsilon \cdot \log(n/\epsilon)^{-1}\big)$. *Additionally, let* $\{S_i\}_{i=1}^{k}$ *be the $k$-way partition corresponding to $\rho(k)$ and suppose that* $\{S_i\}_{i=1}^{k}$ *are almost balanced. Let* $\{A_i\}_{i=1}^{k}$ *be the output of Algorithm 2. With probability at least* $0.9 - \epsilon$, *there exists a permutation* $\sigma : [k] \to [k]$ *such that*

$$\sum_{i=1}^{k} \mathrm{vol}(A_i \triangle S_{\sigma(i)}) = O(\epsilon \cdot \mathrm{vol}(V_G)).$$

*Moreover, the running time of Algorithm 2 is*

$$\widetilde{O}\big(m \cdot \epsilon^{-2}\big) + T_{\mathrm{KM}}(n, k, l),$$

*where $m$ is the number of edges in $G$ and $T_{\mathrm{KM}}(n, k, l)$ is the running time of the $k$-means approximation algorithm on $n$ points in $l$ dimensions.*

**Remark 3.1.** *The assumptions on $\lambda_{k+1}$ and $\rho(k)$ in Theorem 3.1 imply that the graph $G$ can be partitioned into exactly $k$ clusters of low conductance. This is related to previous results which make an assumption on the ratio $\lambda_{k+1}/\rho(k)$ [21, 28, 30]. Macgregor and Sun [21] prove a guarantee like Theorem 3.1 under the assumption that $\lambda_{k+1}/\rho(k) = \Omega(1)$. We achieve a faster algorithm under a slightly stronger assumption.*

**Remark 3.2.** *The running time of Theorem 3.1 improves on previous spectral clustering algorithms. Boutsidis et al. [4] describe an algorithm with running time $\widetilde{O}\big(m \cdot k \cdot \epsilon^{-2}\big) + O\big(k^2 \cdot n\big) + T_{\mathrm{KM}}(n, k, k)$. Moreover, their analysis does not provide any guarantee on the number of misclassified vertices.*

---

**Algorithm 2:** FASTSPECTRALCLUSTER($G = (V, E), k \in \mathbb{Z}_{\geq 0}, \epsilon \in [0, 1]$)

---

1   $\mathbf{M} \leftarrow \mathbf{I} - (1/2) \cdot \mathbf{N}_G$
2   $l \leftarrow \Theta\big(\log(k) \cdot \epsilon^{-2}\big)$
3   $t \leftarrow \Theta\big(\log(n/\epsilon^2 k)\big)$
4   **for** $i \in \{1, \dots, l\}$ **do**
5      Let $\boldsymbol{x}_i \in \mathbb{R}^n$ be a random vector from Gaussian distribution $\mathrm{N}(\mathbf{0}, \mathbf{I})$
6      $\boldsymbol{y}_i \leftarrow \text{POWERMETHOD}(\mathbf{M}, \boldsymbol{x}_i, t)$
7   **end**
8   $\mathbf{Y} \leftarrow [\boldsymbol{y}_1; \dots; \boldsymbol{y}_l]$
9   $A_1, \dots, A_k \leftarrow \text{KMEANS}(\mathbf{D}_G^{-1/2} \mathbf{Y}, k)$
10   **return** $A_1, \dots, A_k$

---

**Remark 3.3.** *The constants in the definition of $l$ and $t$ in Algorithm 2 are based on those in the analysis of Makarychev et al. [23] and this paper. In practice, we find that setting $l = \log(k)$ and $t = 10 \log(n/k)$ works well.*

Throughout the remainder of this section, we will sketch the proof of Theorem 3.1. We assume that $G = (V, E)$ is a graph with $k$ clusters $\{S_i\}_{i=1}^k$ of almost balanced size, $\lambda_{k+1} = \Omega(1)$, and $\rho(k) = O\big(\epsilon \cdot \log(n/\epsilon)^{-1}\big)$.

In order to understand the behaviour of the $k$-means algorithm on the computed vectors, we will analyse the $k$-means cost of a given partition under three different embeddings of the vertices. Let $\boldsymbol{f}_1, \dots, \boldsymbol{f}_k$ be the eigenvectors of $\mathbf{M}$ corresponding to the eigenvalues $\gamma_1, \dots, \gamma_k$ and let $\boldsymbol{y}_1, \dots, \boldsymbol{y}_l$ be the vectors computed in Algorithm 2. We will also consider the vectors $\boldsymbol{z}_1, \dots, \boldsymbol{z}_l$ given by $\boldsymbol{z}_i = \mathbf{P}\boldsymbol{x}_i$, where $\{\boldsymbol{x}_i\}_{i=1}^k$ are the random vectors sampled in Algorithm 2, and $\mathbf{P} = \sum_{i=1}^k \boldsymbol{f}_i \boldsymbol{f}_i^\mathsf{T}$ is the projection onto the space spanned by $\boldsymbol{f}_1, \dots, \boldsymbol{f}_k$. Notice that each $\boldsymbol{z}_i$ is a random vector distributed according to the $k$-dimensional Gaussian distribution. Furthermore, let

$$\mathbf{F} = \begin{bmatrix} | & & | \\ \boldsymbol{f}_1 & \dots & \boldsymbol{f}_k \\ | & & | \end{bmatrix}, \quad \mathbf{Y} = \begin{bmatrix} | & & | \\ \boldsymbol{y}_1 & \dots & \boldsymbol{y}_l \\ | & & | \end{bmatrix} \quad \text{and} \quad \mathbf{Z} = \begin{bmatrix} | & & | \\ \boldsymbol{z}_1 & \dots & \boldsymbol{z}_l \\ | & & | \end{bmatrix}.$$

We will consider the vertex embeddings given by $\mathbf{D}^{-1/2}\mathbf{F}$, $\mathbf{D}^{-1/2}\mathbf{Z}$ and $\mathbf{D}^{-1/2}\mathbf{Y}$ and show that the $k$-means objective for every $k$-way partition is approximately equal in each of them. We will use the following result shown by Makarychev et al. [23].

**Lemma 3.2** ([23], Theorem 1.3). *Given data $\mathbf{X} \in \mathbb{R}^{n \times k}$, let $\mathbf{\Pi} \in \mathbb{R}^{k \times l}$ be a random matrix with each column sampled from the $k$-dimensional Gaussian distribution $\mathrm{N}(\mathbf{0}, \mathbf{I}_k)$ and*

$$l = O\left(\frac{\log(k) + \log(1/\epsilon)}{\epsilon^2}\right).$$

*Then, with probability at least $1 - \epsilon$, it holds for all partitions $\{A_i\}_{i=1}^k$ of $[n]$ that*

$$\text{COST}_\mathbf{X}(A_1, \dots, A_k) \in (1 \pm \epsilon)\text{COST}_{\mathbf{X}\mathbf{\Pi}}(A_1, \dots, A_k).$$

Applying this lemma with $\mathbf{X} = \mathbf{D}^{-\frac{1}{2}}\mathbf{F}$ and $\mathbf{\Pi} = \mathbf{F}^\mathsf{T}\mathbf{Z}$ shows that the $k$-means cost is approximately equal in the embeddings given by $\mathbf{D}^{-\frac{1}{2}}\mathbf{F}$ and $\mathbf{D}^{-\frac{1}{2}}\mathbf{Z}$, since $\mathbf{F}\mathbf{F}^\mathsf{T}\mathbf{Z} = \mathbf{Z}$ and each of the entries of $\mathbf{F}^\mathsf{T}\mathbf{Z}$ is distributed according to the Gaussian distribution $\mathrm{N}(0, 1)$.[2] By Lemma 3.1, we can also show that the $k$-means objective in $\mathbf{D}^{-\frac{1}{2}}\mathbf{Y}$ is within an additive error of $\mathbf{D}^{-\frac{1}{2}}\mathbf{Z}$. This allows us to prove the following lemma.

**Lemma 3.3.** *With probability at least $0.9 - \epsilon$, for any partitioning $\{A_i\}_{i=1}^k$ of the vertex set $V$, we have*

$$\text{COST}_{\mathbf{D}^{-1/2}\mathbf{Y}}(A_1, \dots, A_k) \geq (1 - \epsilon)\text{COST}_{\mathbf{D}^{-1/2}\mathbf{F}}(A_1, \dots, A_k) - \epsilon k$$

*and*

$$\text{COST}_{\mathbf{D}^{-1/2}\mathbf{Y}}(A_1, \dots, A_k) \leq (1 + \epsilon)\text{COST}_{\mathbf{D}^{-1/2}\mathbf{F}}(A_1, \dots, A_k) + \epsilon k.$$

---

[2]There is some interesting subtlety in this argument. If we ignore the $\mathbf{D}^{-\frac{1}{2}}$ matrix, then $\mathbf{F}$ is the data matrix, and $\mathbf{F}^\mathsf{T}\mathbf{Z}$ represents the random projection. After projecting the data, we are left with the projection matrix $\mathbf{Z}$ itself. This happens only because the data matrix is the orthonormal basis $\mathbf{F}$ which is also used to project $\mathbf{Z}$.

To complete the proof of Theorem 3.1, we will make use of the following results proved by Macgregor and Sun [21].

**Lemma 3.4** ([21], Lemma 4.1). *There exists a partition $\{A_i\}_{i=1}^k$ of the vertex set $V$ such that*

$$\text{COST}_{\mathbf{D}^{-1/2}\mathbf{F}}(A_1, \ldots A_k) < k \cdot \rho(k)/\lambda_{k+1}.$$

**Lemma 3.5** ([21], Theorem 2). *Given some partition of the vertices, $\{A_i\}_{i=1}^k$, such that*

$$\text{COST}_{\mathbf{D}^{-1/2}\mathbf{F}}(A_1, \ldots A_k) \leq c \cdot k,$$

*then there exists a permutation $\sigma : [k] \to [k]$ such that*

$$\sum_{i=1}^{k} \text{vol}(A_i \triangle S_{\sigma(i)}) = O(c \cdot \text{vol}(V)).$$

*Proof of Theorem 3.1.* By Lemma 3.4 and Lemma 3.3, with probability at least $0.9 - \epsilon$, there exists some partition $\{\widehat{A}_i\}_{i=1}^k$ of the vertex set $V_G$ such that

$$\text{COST}_{\mathbf{D}^{-1/2}\mathbf{Y}}(\widehat{A}_1, \ldots, \widehat{A}_k) = O\left((1+\epsilon)\frac{\epsilon k}{\log(n/\epsilon)} + \epsilon k\right).$$

Since we use a constant-factor approximation algorithm for $k$-means, the partition $\{A_i\}_{i=1}^k$ returned by Algorithm 2 satisfies $\text{COST}_{\mathbf{D}^{-1/2}\mathbf{Y}}(A_1, \ldots, A_k) = O(\epsilon k)$. Then, by Lemma 3.5 and Lemma 3.3, for some permutation $\sigma : [k] \to [k]$, we have

$$\sum_{i=1}^{k} \text{vol}(A_i \triangle S_{\sigma(i)}) = O(\epsilon \cdot \text{vol}(V_G)).$$

To bound the running time, notice that the number of non-zero entries in $\mathbf{M}$ is $2m$, and the time complexity of matrix multiplication is proportional to the number of non-zero entries. Therefore, the running time of POWERMETHOD($\mathbf{M}, \boldsymbol{x}_0, t$) is $\widetilde{O}(m)$. Since the loop in Algorithm 2 is executed $\Theta(\log(k) \cdot \epsilon^{-2})$ times, the total running time of Algorithm 2 is $\widetilde{O}(m \cdot \epsilon^{-2}) + T_{\text{KM}}(n, k, l)$. ∎

## 4 Experiments

In this section, we empirically study several variants of the spectral clustering algorithm. We compare the following algorithms:

- $k$ EIGENVECTORS: the classical spectral clustering algorithm which uses $k$ eigenvectors of the graph Laplacian matrix to embed the vertices. This is the algorithm analysed in [21, 28].
- $\log(k)$ EIGENVECTORS: spectral clustering with $\log(k)$ eigenvectors of the graph Laplacian to embed the vertices.
- KASP: the fast spectral clustering algorithm proposed by Yan et al. [35]. The algorithm proceeds by first coarsening the data with $k$-means before applying spectral clustering.
- PM $k$ VECTORS (Power Method with $k$ vectors): spectral clustering with $k$ orthogonal vectors computed with the power method. This is the algorithm analysed in [4].
- PM $\log(k)$ VECTORS: spectral clustering with $\log(k)$ random vectors computed with the power method. This is Algorithm 2.

We implement all algorithms in Python, using the `numpy` [12], `scipy` [32], `stag` [22], and `scikit-learn` [27], libraries for matrix manipulation, eigenvector computation, graph processing, and $k$-means approximation respectively. We first compare the performance of the algorithms on synthetic graphs with a range of sizes drawn from the stochastic block model (SBM). We then study the algorithms' performance on several real-world datasets. We find that our algorithm is significantly faster than all other spectral clustering algorithm, while maintaining almost the same clustering accuracy. The most significant improvement is seen on graphs with a large number of clusters. All experiments are performed on an HP laptop with an 11th Gen Intel(R) Core(TM) i7-11800H @ 2.30GHz processor and 32 GB RAM. The code to reproduce the experiments is available at `https://github.com/pmacg/fast-spectral-clustering`.

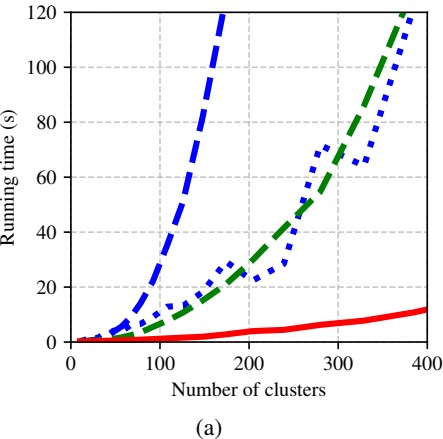 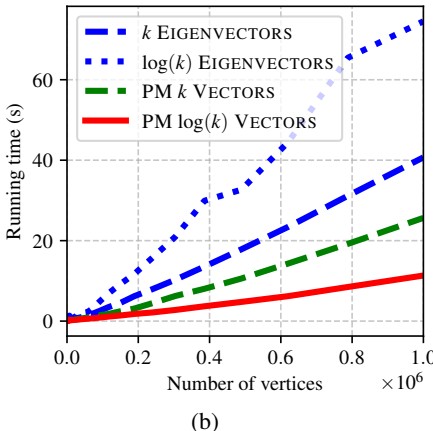

$$(a) \qquad\qquad\qquad (b)$$

Figure 2: The running time of spectral clustering variants on graphs drawn from the stochastic block model. (a) Setting $n = 1000 \cdot k$ and increasing the number of clusters, $k$, shows that Algorithm 2 is much faster than alternative methods for large values of $k$. (b) Setting $k = 20$ and increasing the number of vertices, $n$, shows that for fixed $k$, Algorithm 2 is faster than the alternatives by a constant factor.

## 4.1 Synthetic Data

In this section, we evaluate the spectral clustering algorithms on synthetic data drawn from the stochastic block model. Given parameters $n \in \mathbb{Z}_{\geq 0}$, $k \in \mathbb{Z}_{\geq 0}$, $p \in [0, 1]$, and $q \in [0, 1]$, we generate a graph $G = (V, E)$ with $n$ vertices and $k$ ground-truth clusters $S_1, \ldots, S_k$ of size $n/k$. For any pair of vertices $u \in S_i$ and $v \in S_j$, we add the edge $\{u, v\}$ with probability $p$ if $i = j$ and with probability $q$ otherwise. We study the running time of the algorithms in two settings.

In the first experiment, we set $n = 1000 \cdot k$, $p = 0.04$, and $q = 1/(1000k)$. Then, we study the running time of spectral clustering for different values of $k$. The results are shown in Figure 2(a). We observe that our newly proposed algorithm is much faster than existing methods for large values of $k$, and our algorithm is easily able to scale to large graphs with several hundred thousand vertices.

In the second experiment, we set $k = 20$, $p = 40/n$, and $q = 1/(20n)$. Then, we study the running time of the spectral clustering algorithms for different values of $n$. The results are shown in Figure 2(b). Empirically, we find that when $k$ is a fixed constant, our newly proposed algorithm is faster than existing methods by a constant factor. In every case, all algorithms successfully recover the ground truth clusters.[3]

## 4.2 Real-world Data

In this section, we evaluate spectral clustering on real-world data with labeled ground-truth clusters. We compare the algorithms on the following datasets from a variety of domains.

- **MNIST** [18]: each data point is an image with $28 \times 28$ greyscale pixels, representing a hand-written digit from 0 to 9.

- **Pen Digits** [1]: data is collected by writing digits on a digital pen tablet. Each data point corresponds to some digit from 0 to 9 and consists of 8 pairs of $(x, y)$ coordinates encoding the sequence of pen positions while the digit was written.

- **Fashion** [34]: each data point is an image with $28 \times 28$ greyscale pixels, representing one of 10 classes of fashion item, such as 'shoe' or 'shirt'.

- **HAR** (Human Activity Recognition) [2]: the dataset consists of pre-processed sensor data from a body-worn smartphone. Participants were asked to perform a variety of activities,

---

[3]Note that we cannot compare with the KASP algorithm on the stochastic block model since KASP is designed to operate on vector data rather than on graphs.

Table 1: The performance of spectral clustering algorithms on real-world datasets. The PM $\log(k)$ algorithm corresponds to Algorithm 2. We perform 10 trials and report the average performance with one standard deviation of uncertainty. We observe that Algorithm 2 is consistently very fast when compared to the other algorithms while achieving comparable clustering accuracy.

| | | Dataset | | | | |
|---|---|---|---|---|---|---|
| | Algorithm | MNIST | Pen Digits | Fashion | HAR | Letter |
| Time | $k$ EIGS | $2.70 \pm 0.24$ | $0.64 \pm 0.07$ | $3.55 \pm 0.17$ | $0.58 \pm 0.07$ | $29.29 \pm 11.85$ |
| | $\log(k)$ EIGS | $2.73 \pm 0.20$ | $1.01 \pm 0.05$ | $3.79 \pm 0.11$ | $0.83 \pm 0.05$ | $24.99 \pm 11.58$ |
| | KASP | $15.47 \pm 3.40$ | $\mathbf{0.22 \pm 0.03}$ | $14.33 \pm 3.54$ | $0.91 \pm 0.19$ | $\mathbf{0.33 \pm 0.14}$ |
| | PM $k$ | $3.23 \pm 0.15$ | $0.49 \pm 0.02$ | $2.40 \pm 0.07$ | $0.38 \pm 0.01$ | $1.14 \pm 0.02$ |
| | PM $\log(k)$ | $\mathbf{1.99 \pm 0.05}$ | $0.36 \pm 0.01$ | $\mathbf{1.19 \pm 0.06}$ | $\mathbf{0.30 \pm 0.02}$ | $0.39 \pm 0.02$ |
| ARI | $k$ EIGS | $\mathbf{0.61 \pm 0.01}$ | $\mathbf{0.58 \pm 0.02}$ | $\mathbf{0.42 \pm 0.00}$ | $\mathbf{0.51 \pm 0.00}$ | $\mathbf{0.17 \pm 0.00}$ |
| | $\log(k)$ EIGS | $0.49 \pm 0.03$ | $\mathbf{0.63 \pm 0.06}$ | $0.32 \pm 0.02$ | $0.30 \pm 0.01$ | $\mathbf{0.17 \pm 0.00}$ |
| | KASP | $0.33 \pm 0.03$ | $0.42 \pm 0.04$ | $0.30 \pm 0.03$ | $\mathbf{0.48 \pm 0.03}$ | $0.13 \pm 0.01$ |
| | PM $k$ | $0.55 \pm 0.03$ | $\mathbf{0.60 \pm 0.07}$ | $0.40 \pm 0.02$ | $0.50 \pm 0.02$ | $\mathbf{0.17 \pm 0.00}$ |
| | PM $\log(k)$ | $0.51 \pm 0.04$ | $\mathbf{0.61 \pm 0.05}$ | $0.35 \pm 0.03$ | $0.49 \pm 0.02$ | $\mathbf{0.17 \pm 0.00}$ |
| NMI | $k$ EIGS | $\mathbf{0.74 \pm 0.01}$ | $\mathbf{0.78 \pm 0.00}$ | $\mathbf{0.60 \pm 0.00}$ | $\mathbf{0.72 \pm 0.00}$ | $0.27 \pm 0.02$ |
| | $\log(k)$ EIGS | $0.68 \pm 0.01$ | $\mathbf{0.77 \pm 0.03}$ | $0.55 \pm 0.01$ | $0.48 \pm 0.02$ | $0.13 \pm 0.02$ |
| | KASP | $0.48 \pm 0.02$ | $0.60 \pm 0.03$ | $0.46 \pm 0.03$ | $0.61 \pm 0.03$ | $\mathbf{0.35 \pm 0.01}$ |
| | PM $k$ | $\mathbf{0.73 \pm 0.03}$ | $0.76 \pm 0.03$ | $\mathbf{0.61 \pm 0.02}$ | $0.69 \pm 0.02$ | $0.29 \pm 0.03$ |
| | PM $\log(k)$ | $0.69 \pm 0.02$ | $\mathbf{0.77 \pm 0.02}$ | $\mathbf{0.55 \pm 0.04}$ | $0.66 \pm 0.04$ | $0.30 \pm 0.01$ |

such as 'walking', 'walking upstairs', and 'standing'. The task is to identify the activity from the sensor data.

- **Letter** [9]: each data point corresponds to an upper-case letter from 'A' to 'Z'. The data was generated from distorted images of letters with a variety of fonts, and the features correspond to various statistics computed on the resulting images.

The datasets are all made available by the OpenML [31] project, and can be downloaded with the `scikit-learn` library [27]. We first pre-process each dataset by computing the $k$ nearest neighbour graph from the data, for $k = 10$. Table 2 shows the number of nodes and the number of ground truth clusters in each dataset.

For each dataset, we report the performance of each spectral clustering algorithm with respect to the running time in seconds, and the clustering accuracy measured with the Adjusted Rand Index (ARI) [10, 29] and the Normalised Mutual Information (NMI) [17]. Table 1 summarises the results.

Table 2: The number of vertices ($n$) and clusters ($k$) in each of the real-world datasets.

| Dataset | $n$ | $k$ |
|---|---|---|
| MNIST | 70000 | 10 |
| Pen Digits | 7494 | 10 |
| Fashion | 70000 | 10 |
| HAR | 10299 | 6 |
| Letter | 20000 | 26 |

We find that Algorithm 2 is consistently very fast when compared to the other spectral clustering algorithms. Moreover, the clustering accuracy is similar for every algorithm.

## 5 Conclusion

In this paper, we introduced a new fast spectral clustering algorithm based on projecting the vertices of the graph into $O(\log(k))$ dimensions with the power method. We find that the new algorithm is faster than previous spectral clustering algorithms and achieves similar clustering accuracy.

This algorithm offers a new option for the application of spectral clustering. If a large running time is acceptable and the goal is to achieve the best accuracy possible, then our experimental results suggest that the classical spectral clustering algorithm with $k$ eigenvectors is the optimal choice. On the other hand, when the number of clusters or the number of data points is very large, our newly proposed method provides a significantly faster algorithm for a small trade-off in terms of clustering accuracy. This could allow spectral clustering to be applied in regimes that were previously intractable, such as when $k = \Theta(n)$.

## Acknowledgements

This work is supported by EPSRC Early Career Fellowship (EP/T00729X/1).

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

# A Omitted detail from Section 3

In this section, we prove the main theoretical result of the paper. In order that this section is self-contained, we repeat some of the steps included in the main paper. We first show that the lengths of the random vectors $\boldsymbol{x}_i$ generated in Algorithm 2 are close to their expected value. Notice that $\mathrm{E}\left[\|\boldsymbol{x}_i\|_2\right] = \sqrt{n}$ and $\mathrm{E}\left[\|\mathbf{P}\boldsymbol{x}_i\|_2\right] = \sqrt{k}$. We use Chebyshev's inequality to show the following.

**Lemma A.1.** *Let $\boldsymbol{x} \in \mathbb{R}^n$ be drawn from the $n$-dimensional Gaussian distribution $\mathrm{N}(\mathbf{0}, \mathbf{I})$. Let $\boldsymbol{f}_1, \ldots, \boldsymbol{f}_k \in \mathbb{R}^n$ be orthogonal vectors and let $\mathbf{P} = \sum_{i=1}^{k} \boldsymbol{f}_i \boldsymbol{f}_i^\intercal$ be the projection onto the space spanned by $\boldsymbol{f}_1, \ldots, \boldsymbol{f}_k$. With probability at least $1 - (1/10k)$,*

- *$\|\mathbf{P}\boldsymbol{x}\|_2 \leq \sqrt{6k}$, and*

- *$\|\boldsymbol{x}\|_2 \leq \sqrt{6n}$.*

*Proof of Lemma A.1.* Since $\boldsymbol{x}$ is drawn from a symmetric $n$-dimensional Gaussian distribution, $\|\boldsymbol{x}\|_2^2$ is distributed according to a $\chi^2$ distribution with $n$ degrees of freedom. Similarly, since $\mathbf{P}$ is a projection matrix, $\|\mathbf{P}\boldsymbol{x}\|_2^2$ is distributed according to a $\chi^2$ distribution with $k$ degrees of freedom. By the Chebyshev inequality, we have that

$$\Pr\left[\|\mathbf{P}\boldsymbol{x}_i\|_2^2 \geq 6k\right] \leq \frac{k}{(5k)^2} = \frac{1}{25k},$$

and

$$\Pr\left[\|\boldsymbol{x}_i\|_2^2 \geq 6n\right] \leq \frac{n}{(5n)^2} = \frac{1}{25n}.$$

The lemma follows by the union bound and since $k \leq n$. ■

We now show that the output of the POWERMETHOD algorithm is close to a random vector in the space spanned by $\boldsymbol{f}_1, \ldots, \boldsymbol{f}_k$.

**Lemma 3.1.** *Let $\mathbf{M} \in \mathbb{R}^{n \times n}$ be a matrix with eigenvalues $1 \geq \gamma_1 \geq \ldots \geq \gamma_n \geq 0$ and corresponding eigenvectors $\boldsymbol{f}_1, \ldots \boldsymbol{f}_n$. Let $\boldsymbol{x}_0 \in \mathbb{R}^n$ be drawn from the $n$-dimensional Gaussian distribution $N(\mathbf{0}, \mathbf{I})$. Let $\boldsymbol{x}_t = \text{POWERMETHOD}(\mathbf{M}, \boldsymbol{x}_0, t)$ for $t = \Theta\left(\log(n/\epsilon^2 k)\right)$. If $\gamma_k \geq 1 - O\left(\epsilon \cdot \log(n/\epsilon^2 k)^{-1}\right)$ and $\gamma_{k+1} \leq 1 - \Omega(1)$, then with probability at least $1 - 1/(10k)$,*

$$\|\boldsymbol{x}_t - \mathbf{P}\boldsymbol{x}_0\|_2 \leq \epsilon\sqrt{k},$$

*where $\mathbf{P} = \sum_{i=1}^{k} \boldsymbol{f}_i \boldsymbol{f}_i^\intercal$ is the projection onto the space spanned by the first $k$ eigenvectors of $\mathbf{M}$.*

*Proof of Lemma 3.1.* By the assumptions of the Lemma, we can assume that

- $\gamma_{k+1} \leq c_1 < 1$,
- $\gamma_k \geq 1 - c_2\epsilon \log(24n/\epsilon^2 k)^{-1}$, and
- $t = c_3 \log(24n/\epsilon^2 k)$,

for constants $c_1$, $c_2$, and $c_3$. Fixing $c_1 < 1$, we will set

$$c_3 = \frac{1}{2\log\left(\frac{1}{c_1}\right)}$$

and

$$c_2 = \frac{1}{c_3 \cdot 2\sqrt{6}}.$$

Furthermore, by Lemma A.1, with probability at least $1 - (1/10k)$ it holds that

$$\|\mathbf{P}\boldsymbol{x}_0\|_2 \leq \sqrt{6k}$$

and
$$\|\boldsymbol{x}_0\|_2 \le \sqrt{6n},$$
and we assume that this holds in the remainder of the proof.

Now, we write $\boldsymbol{x}_0$ in terms of its expansion in the basis given by the eigenvectors $\boldsymbol{f}_1, \ldots, \boldsymbol{f}_n$:
$$\boldsymbol{x}_0 = \sum_{j=1}^{n} a_j \boldsymbol{f}_j,$$
where $a_j = \langle \boldsymbol{x}_0, \boldsymbol{f}_j \rangle$. Similarly, we have
$$\mathbf{P}\boldsymbol{x}_0 = \sum_{j=1}^{k} a_j \boldsymbol{f}_j$$
and
$$\boldsymbol{x}_t = \sum_{j=1}^{n} a_j \gamma_j^t \boldsymbol{f}_j.$$
Then,
$$\begin{aligned}
\|\boldsymbol{x}_t - \mathbf{P}\boldsymbol{x}_0\|_2 &= \left\| \sum_{j=1}^{k} \left(a_j \gamma_j^t - a_j\right) \boldsymbol{f}_j + \sum_{j=k+1}^{n} a_j \gamma_j^t \boldsymbol{f}_j \right\|_2 \\
&\le \left\| \sum_{j=1}^{k} \left(a_j \gamma_j^t - a_j\right) \boldsymbol{f}_j \right\|_2 + \left\| \sum_{j=k+1}^{n} a_j \gamma_j^t \boldsymbol{f}_j \right\|_2 \\
&\le \left\| \left(1 - \gamma_k^t\right) \sum_{j=1}^{k} a_j \boldsymbol{f}_j \right\|_2 + \left\| \gamma_{k+1}^t \sum_{j=k+1}^{n} a_j \boldsymbol{f}_j \right\|_2 \\
&= \left(1 - \gamma_k^t\right) \|\mathbf{P}\boldsymbol{x}_0\|_2 + \gamma_{k+1}^t \|(\mathbf{I} - \mathbf{P})\boldsymbol{x}_0\|_2 \\
&\le \left(1 - \gamma_k^t\right) \|\mathbf{P}\boldsymbol{x}_0\|_2 + \gamma_{k+1}^t \|\boldsymbol{x}_0\|_2
\end{aligned}$$
where we used the fact that $1 \ge \gamma_1 \ge \ldots \ge \gamma_n$. Now, we have
$$\begin{aligned}
\gamma_k^t &\ge \left(1 - c_2 \epsilon \log(24n/\epsilon^2 k)^{-1}\right)^{c_3 \log(24n/\epsilon^2 k)} \\
&\ge 1 - c_2 c_3 \epsilon \\
&= 1 - \frac{\epsilon}{2\sqrt{6}}.
\end{aligned}$$
Furthermore,
$$\begin{aligned}
\gamma_{k+1}^t &\le c_1^{c_3 \log(24n/\epsilon^2 k)} \\
&= \left(\frac{1}{c_1}\right)^{c_3 \log(\epsilon^2 k/24n)} \\
&= \left(\frac{\epsilon^2 k}{24n}\right)^{c_3 \log(1/c_1)} \\
&= \epsilon \sqrt{\frac{k}{24n}}.
\end{aligned}$$
Combining everything together, we have
$$\begin{aligned}
\|\boldsymbol{x}_t - \mathbf{P}\boldsymbol{x}_0\|_2 &\le \frac{\epsilon}{2\sqrt{6}} \|\mathbf{P}\boldsymbol{x}_0\|_2 + \epsilon \sqrt{\frac{k}{24n}} \|\boldsymbol{x}_0\|_2 \\
&\le \frac{\epsilon}{2\sqrt{6}} \sqrt{6k} + \epsilon \sqrt{\frac{6kn}{24n}} \\
&\le \epsilon \sqrt{k},
\end{aligned}$$
which completes the proof. $\blacksquare$

It remains to prove that the $k$-means cost is preserved in the embedding produced by the power method. Recall that $\boldsymbol{f}_1, \ldots, \boldsymbol{f}_k$ are the eigenvectors of $\mathbf{M}$ corresponding to the eigenvalues $\gamma_1, \ldots, \gamma_k$ and $\boldsymbol{y}_1, \ldots, \boldsymbol{y}_l$ are the vectors computed in Algorithm 2. We will also consider the vectors $\boldsymbol{z}_1, \ldots, \boldsymbol{z}_l$ given by $\boldsymbol{z}_i = \mathbf{P}\boldsymbol{x}_i$, where $\{\boldsymbol{x}_i\}_{i=1}^k$ are the random vectors sampled in Algorithm 2, and $\mathbf{P} = \sum_{i=1}^k \boldsymbol{f}_i \boldsymbol{f}_i^\mathsf{T}$ is the projection onto the space spanned by $\boldsymbol{f}_1, \ldots, \boldsymbol{f}_k$. We also define

$$\mathbf{F} = \begin{bmatrix} | & & | \\ \boldsymbol{f}_1 & \cdots & \boldsymbol{f}_k \\ | & & | \end{bmatrix}, \quad \mathbf{Y} = \begin{bmatrix} | & & | \\ \boldsymbol{y}_1 & \cdots & \boldsymbol{y}_l \\ | & & | \end{bmatrix} \quad \text{and} \quad \mathbf{Z} = \begin{bmatrix} | & & | \\ \boldsymbol{z}_1 & \cdots & \boldsymbol{z}_l \\ | & & | \end{bmatrix}.$$

We will use the following result shown by Makarychev et al. [23].

**Lemma 3.2** ([23], Theorem 1.3). *Given data $\mathbf{X} \in \mathbb{R}^{n \times k}$, let $\mathbf{\Pi} \in \mathbb{R}^{k \times l}$ be a random matrix with each column sampled from the $k$-dimensional Gaussian distribution $\mathrm{N}(\mathbf{0}, \mathbf{I}_k)$ and*

$$l = O\left(\frac{\log(k) + \log(1/\epsilon)}{\epsilon^2}\right).$$

*Then, with probability at least $1 - \epsilon$, it holds for all partitions $\{A_i\}_{i=1}^k$ of $[n]$ that*

$$\mathrm{COST}_\mathbf{X}(A_1, \ldots, A_k) \in (1 \pm \epsilon)\mathrm{COST}_{\mathbf{X}\mathbf{\Pi}}(A_1, \ldots, A_k).$$

Applying this lemma with $\mathbf{X} = \mathbf{D}^{-\frac{1}{2}}\mathbf{F}$ and $\mathbf{\Pi} = \mathbf{F}^\mathsf{T}\mathbf{Z}$ shows that the $k$-means cost is approximately equal in the embeddings given by $\mathbf{D}^{-\frac{1}{2}}\mathbf{F}$ and $\mathbf{D}^{-\frac{1}{2}}\mathbf{Z}$, since $\mathbf{F}\mathbf{F}^\mathsf{T}\mathbf{Z} = \mathbf{Z}$ and each of the entries of $\mathbf{F}^\mathsf{T}\mathbf{Z}$ is distributed according to the Gaussian distribution $\mathrm{N}(0, 1)$. By Lemma 3.1, we can also show that the $k$-means objective in $\mathbf{D}^{-\frac{1}{2}}\mathbf{Y}$ is within an additive error of $\mathbf{D}^{-\frac{1}{2}}\mathbf{Z}$. This allows us to prove the following lemma.

**Lemma 3.3.** *With probability at least $0.9 - \epsilon$, for any partitioning $\{A_i\}_{i=1}^k$ of the vertex set $V$, we have*

$$\mathrm{COST}_{\mathbf{D}^{-1/2}\mathbf{Y}}(A_1, \ldots, A_k) \geq (1 - \epsilon)\mathrm{COST}_{\mathbf{D}^{-1/2}\mathbf{F}}(A_1, \ldots, A_k) - \epsilon k$$

*and*

$$\mathrm{COST}_{\mathbf{D}^{-1/2}\mathbf{Y}}(A_1, \ldots, A_k) \leq (1 + \epsilon)\mathrm{COST}_{\mathbf{D}^{-1/2}\mathbf{F}}(A_1, \ldots, A_k) + \epsilon k.$$

In order to prove this, we will use the fact shown by Boutsidis and Magdon-Ismail [5] that we can write the $k$-means cost as

$$\mathrm{COST}_\mathbf{B}(A_i, \ldots, A_k) = \|\mathbf{B} - \mathbf{X}\mathbf{X}^\mathsf{T}\mathbf{B}\|_F^2 \tag{1}$$

where $\mathbf{X} \in \mathbb{R}^{n \times k}$ is the indicator matrix of the partition, defined by

$$\mathbf{X}(u, i) = \begin{cases} \frac{1}{\sqrt{|A_i|}} & \text{if } u \in A_i \\ 0 & \text{otherwise} \end{cases},$$

and $\|\mathbf{B}\|_F \triangleq (\sum_{i,j} \mathbf{B}_{i,j}^2)^{1/2}$ is the Frobenius norm.

*Proof of Lemma 3.3.* Notice that $\mathbf{F}^\mathsf{T}\mathbf{Z} \in \mathbb{R}^{k \times l}$ is a random matrix with columns drawn from the standard $k$-dimensional Gaussian distribution. Then, by Lemma 3.2, with probability at least $1 - \epsilon$, we have for any partition $\{A_i\}_{i=1}^k$ that

$$\mathrm{COST}_{\mathbf{D}^{-1/2}\mathbf{F}}(A_1, \ldots, A_k) \in (1 \pm \epsilon)\,\mathrm{COST}_{\mathbf{D}^{-1/2}\mathbf{Z}}(A_1, \ldots, A_k) \tag{2}$$

since $\mathbf{D}^{-1/2}\mathbf{F}\mathbf{F}^\mathsf{T}\mathbf{Z} = \mathbf{D}^{-1/2}\mathbf{Z}$, where we use the fact that the columns of $\mathbf{Z}$ are in the span of $\boldsymbol{f}_1, \ldots, \boldsymbol{f}_k$.

Furthermore, by the union bound, we can assume with probability at least 0.9 that the conclusion of Lemma 3.1 holds for every vector $\boldsymbol{y}_i$ computed by Algorithm 2.

Now, we will establish that $\mathrm{COST}_{\mathbf{D}^{-1/2}\mathbf{Y}}(\cdot)$ is close to $\mathrm{COST}_{\mathbf{D}^{-1/2}\mathbf{Z}}(\cdot)$ which will complete the proof. For some arbitrary partition $\{A_i\}_{i=1}^k$, let $\mathbf{X}$ be the indicator matrix of the partition. Then, we

have

$$\left\|\mathbf{D}^{-1/2}\mathbf{Y} - \mathbf{X}\mathbf{X}^{\mathsf{T}}\mathbf{D}^{-1/2}\mathbf{Y}\right\|_F - \left\|\mathbf{D}^{-1/2}\mathbf{Z} - \mathbf{X}\mathbf{X}^{\mathsf{T}}\mathbf{D}^{-1/2}\mathbf{Z}\right\|_F$$

$$= \left\|\left(\mathbf{I} - \mathbf{X}\mathbf{X}^{\mathsf{T}}\right)\mathbf{D}^{-1/2}\mathbf{Y}\right\|_F - \left\|\left(\mathbf{I} - \mathbf{X}\mathbf{X}^{\mathsf{T}}\right)\mathbf{D}^{-1/2}\mathbf{Z}\right\|_F$$

$$\leq \left\|\left(\mathbf{I} - \mathbf{X}\mathbf{X}^{\mathsf{T}}\right)\mathbf{D}^{-1/2}\left(\mathbf{Y} - \mathbf{Z}\right)\right\|_F$$

$$\leq \left\|\left(\mathbf{I} - \mathbf{X}\mathbf{X}^{\mathsf{T}}\right)\left(\mathbf{Y} - \mathbf{Z}\right)\right\|_F$$

$$\leq \left\|\mathbf{Y} - \mathbf{Z}\right\|_F$$

$$= \sqrt{\sum_{i=1}^{l} \|\boldsymbol{y}_i - \boldsymbol{z}_i\|_2^2}$$

$$\leq \sqrt{l\epsilon^2 k}$$

$$\leq \epsilon k,$$

Where we use Lemma 3.1, and the fact that $l \leq k$. Combining this with (2) completes the proof. ∎

Now we come to the proof of the main theorem.

**Theorem 3.1.** *Let $G$ be a graph with $\lambda_{k+1} = \Omega(1)$ and $\rho(k) = O\big(\epsilon \cdot \log(n/\epsilon)^{-1}\big)$. Additionally, let $\{S_i\}_{i=1}^{k}$ be the $k$-way partition corresponding to $\rho(k)$ and suppose that $\{S_i\}_{i=1}^{k}$ are almost balanced. Let $\{A_i\}_{i=1}^{k}$ be the output of Algorithm 2. With probability at least $0.9 - \epsilon$, there exists a permutation $\sigma : [k] \to [k]$ such that*

$$\sum_{i=1}^{k} \mathrm{vol}(A_i \triangle S_{\sigma(i)}) = O(\epsilon \cdot \mathrm{vol}(V_G)).$$

*Moreover, the running time of Algorithm 2 is*

$$\widetilde{O}\big(m \cdot \epsilon^{-2}\big) + T_{\mathrm{KM}}(n, k, l),$$

*where $m$ is the number of edges in $G$ and $T_{\mathrm{KM}}(n, k, l)$ is the running time of the $k$-means approximation algorithm on $n$ points in $l$ dimensions.*

To complete the proof, we will make use of the following results proved by Macgregor and Sun [21], which hold under the same assumptions as Theorem 3.1.

**Lemma 3.4** ([21], Lemma 4.1)**.** *There exists a partition $\{A_i\}_{i=1}^{k}$ of the vertex set $V$ such that*

$$\mathrm{COST}_{\mathbf{D}^{-1/2}\mathbf{F}}(A_1, \dots A_k) < k \cdot \rho(k)/\lambda_{k+1}.$$

**Lemma 3.5** ([21], Theorem 2)**.** *Given some partition of the vertices, $\{A_i\}_{i=1}^{k}$, such that*

$$\mathrm{COST}_{\mathbf{D}^{-1/2}\mathbf{F}}(A_1, \dots A_k) \leq c \cdot k,$$

*then there exists a permutation $\sigma : [k] \to [k]$ such that*

$$\sum_{i=1}^{k} \mathrm{vol}(A_i \triangle S_{\sigma(i)}) = O(c \cdot \mathrm{vol}(V)).$$

*Proof of Theorem 3.1.* By Lemma 3.4 and Lemma 3.3, with probability at least $0.9 - \epsilon$, there exists some partition $\{\widehat{A}_i\}_{i=1}^{k}$ of the vertex set $V_G$ such that

$$\mathrm{COST}_{\mathbf{D}^{-1/2}\mathbf{Y}}(\widehat{A}_1, \dots, \widehat{A}_k) = O\bigg((1+\epsilon)\frac{\epsilon k}{\log(n/\epsilon)} + \epsilon k\bigg).$$

Since we use a constant-factor approximation algorithm for $k$-means, the partition $\{A_i\}_{i=1}^{k}$ returned by Algorithm 2 satisfies $\mathrm{COST}_{\mathbf{D}^{-1/2}\mathbf{Y}}(A_1, \dots, A_k) = O(\epsilon k)$. Then, by Lemma 3.5 and Lemma 3.3, for some permutation $\sigma : [k] \to [k]$, we have

$$\sum_{i=1}^{k} \mathrm{vol}(A_i \triangle S_{\sigma(i)}) = O(\epsilon \cdot \mathrm{vol}(V_G)).$$

To bound the running time, notice that the number of non-zero entries in $\mathbf{M}$ is $2m$, and the time complexity of matrix multiplication is proportional to the number of non-zero entries. Therefore, the running time of POWERMETHOD$(\mathbf{M}, \boldsymbol{x}_0, t)$ is $\widetilde{O}(m)$. Since the loop in Algorithm 2 is executed $\Theta\big(\log(k) \cdot \epsilon^{-2}\big)$ times, the total running time of Algorithm 2 is $\widetilde{O}\big(m \cdot \epsilon^{-2}\big) + T_{\mathrm{KM}}(n, k, l)$. $\blacksquare$

