# OpenReview forum: "Fast and Simple Spectral Clustering in Theory and Practice"
_NeurIPS.cc/2023/Conference — NeurIPS 2023 poster_

### Official Review · Reviewer_6vxh · 2023-06-30

**Soundness:** 4 excellent
**Presentation:** 4 excellent
**Contribution:** 4 excellent
**Rating:** 7
**Confidence:** 2

**Summary:**

The paper is concerned with the spectral method for clustering vertices of graph. Specifically, the authors find that the classical spectral clustering methods need eigendecomposition of the graph Laplacian matrix which is computationally expensive, and that the authors propose to use a power method to bypass this computation-demanding part. The essence is that the power method is a random projection of the eigendecomposition into subembeddings.

In general, I do find that the idea of the paper is novel, and that the theoretical and experimental results are convincing.

**Strengths:**

The proposed algorithm is very simple to understand and implement. Moreover, the authors have provided both theoretical and experimental results to prove that the algorithm is valid and novel.

The structure of the paper is engaging and is written easy to follow.

**Weaknesses:**

The experiments only compared to the classical eigendecomposition-based methods. It is true that the demanding computation of the embedding step motivates the authors to use a power method for the random project, however, I am not convinced that the eigendecomposition is the only method to do the graph clustering. The authors should add both empirical discussion and experiments to other such related methods.

**Questions:**

The authors claim that the codes are released. however, I don't see them anywhere mentioned in the paper.

pp. 3. l91. The definition of vol is not clear.

pp. 5. l132. Definition of the inner product is not clear. Is it a dot product, or E[<.,.>], since you have random variable?

pp. 5. l142. Is it 1/100k or 1/(100k)?

pp. 5. l143. Should it be bold face P?

**Limitations:**

No. The authors did not discuss the limitations of their method.

---

> ### Author Rebuttal · Authors · 2023-08-09
>
> Thank you for your positive review. We address your questions below.
>
> **Code.**
> The code to reproduce our experiments is included with the submission in the supplementary material.
> When the paper is published, we will make the code publicly available online.
>
> **Editorial points.**
> Thank you for your editorial points, we will update the paper based on your feedback.
> On line 132 it is a dot product not an expectation, and line 142 should be $1 / (100k)$ - we will make this clearer.

---

> > ### Comment · Reviewer_6vxh · 2023-08-16
> >
> > Thank you for your responses. After reading yours and other reviewers' comments, I will keep my score unchanged.

---

### Official Review · Reviewer_6nQB · 2023-07-04

**Soundness:** 4 excellent
**Presentation:** 3 good
**Contribution:** 3 good
**Rating:** 4
**Confidence:** 5

**Summary:**

This paper proposed a new spectral-based graph clustering algorithm. The main novelty of this algorithm is a fast implementation of vertex embedding, built on the power method.

Previous algorithms achieved a low dimensional embedding via two steps:
1) compute k eigenvectors of the graph Laplacian matrix;
2) embed k dimension datapoint into (log k)-dimension via a random projection.

Under this approach, the first step may cost a lot of time, i.e., $O(kn^2)$ in a worst-case analysis. This paper simplifies this step with a power method. Concretely, their algorithm starts with (log k) random vectors $x_1,\dots,x_r$, and calculates $M^{t} x_{i}$. The authors showed that the vectors obtained in this way can approximate random vectors from the eigenspace of $M$. For a sparse matrix $M$, this approach can be implemented much faster. Specifically, if $M$ is a graph matrix, this algorithm can be implemented in time linear in the size of $G$.

Power methods are a very well known approach to approximate eigenvectors. We can usually approximate the first eigenvector very fast for sparse matrices. However, for the second eigenvector, the running time might be slower since the matrix is no longer sparse after removing the largest eigenvector. The beauty of this paper is that we can approximate all $k$ eigenvectors in one shot in some settings. However, there is no free lunch. The main theorem of this paper requires the size of clusters should be balanced. I think this assumption plays a crucial role in this algorithm.

**Strengths:**

1. This paper provides a nice idea to approximate the eigenvectors of the graph Laplacian matrix. The running time of this algorithm is much faster than the previous algorithms.

2. The theoretical analysis is solid and interesting!

3. The power method has good potential. Beyond approximating top eigenvectors, it might have applications in different steps of clustering.

**Weaknesses:**

I first list some minor cons in this paper.

1. It requires the knowledge of k

2. The parameter q is too small in the SBM simulation.

3. The authors used some subjective words to describe their algorithm, such as they call their algorithm a simple algorithm, and their assumptions in the main theorem are natural assumptions. I disagree with the second statement and find these assumptions to be not natural at all. However, I agree that this is a subjective viewpoint and thus put this as a minor con.

4. In the real dataset experiments, it seems that none of the four algorithms are very exciting. For example, the highest NMI achieved for MNIST is 0.74 by spectral clustering through k eigenvectors. However, a quick online search shows that there are many  algorithms with NMI higher than 0.9. See the link: https://paperswithcode.com/sota/image-clustering-on-mnist-full

5. My main concern in this paper is the balancedness assumption. This assumption is not true in many applications. In addition, I think this assumption is crucial for the performance of the algorithm (and not just the analysis), meaning this is an inherent barrier. If the authors want to make this algorithm useful in practice, I think this is a must-be-addressed issue. I am happy to increase my evaluation if the authors can address this barrier. Let me ask a concrete question. Let’s consider the following SBM setting with p = 0.01 and q=0.00001. We assume there is a large cluster of size n/2, and the rest of the clusters have size n^{0.99}/2. Could this power method-based algorithm recover these clusters? Regarding the experiments part, I know that MNIST and Pen Digits are balanced datasets. Are there any unbalanced datasets in the experiments? The authors did not mention the size of each cluster in the paper.

**Questions:**

Could the authors address my concerns about the unbalanced cases?

**Limitations:**

I have included my suggestions in the weakness part.

---

> ### Author Rebuttal · Authors · 2023-08-09
>
> Many thanks for your detailed and thoughtful review and questions.
>
> **Balancedness assumption.**
> We address the balancedness assumption in our top-level comment.
> Furthermore, in the attached PDF, we include the results of the requested SBM experiment.
> We set $p = 0.01$ and $q = 0.00001$ as suggested, and let one cluster have size $n / 2$ and the others have size $n^{0.7} / 2$.
> Note that we use a smaller power of $n$ than suggested in order that the number of small clusters grows faster.
> We find that our algorithm is able to successfully recover the imbalanced clusters under this regime.
>
> Thank you for raising this question, and for the concrete suggestion. We will add some discussion on the imbalanced case to the next version of the paper, and include the additional experiment.
> We hope that you will consider increasing your evaluation based on this and we are happy to continue the discussion if you have further questions.
>
> **Weakness 1.**
> Note that this is a weakness of spectral clustering in general.
> In fact, our algorithm is less susceptible to this weakness than previous spectral clustering algorithms since it is easier to 'guess' a value for $\log(k)$ - in practice, you can often just use a small constant.
> Then well-established heuristics for picking $k$ for the $k$-means step can be applied.
>
> **Weakness 2.**
> We have chosen $q$ in the SBM experiments such that the expected conductance of the generated clusters remain constant while $k$ grows.
>
> **Weakness 3.**
> We understand your point about the use of the phrase 'natural assumptions'. We follow many previous works in the graph clustering literature which make assumptions on the value of $\lambda_{k+1}$ and $\rho(k)$ [15, 19, 21, 23, 27].
>
> **Weakness 4.**
> We agree that spectral clustering does not approach the state of the art on these datasets, and we cannot claim to compete with supervised methods.
>
> **Weakness 5.** See our answer above.

---

> > ### Comment · Reviewer_6nQB · 2023-08-10
> >
> > Thank you for the response.
> >
> > I want to note that the authors have defended their weakness (weakness 1, 3,4) by emphasizing that previous spectral algorithms also shared similar weaknesses.
> >
> > Regards to Weakness 4: I know it is unfair to compare supervised and unsupervised algorithms. However, in that list, there are some unsupervised methods such as *Tree-SNE: Hierarchical Clustering and Visualization Using t-SNE*. As a practitioner, in my observation, some recent manifold-based *unsupervised* clustering algorithms have really good performance.
> >
> > Regards to Weakness 5. I didn't make my question clear. Sorry about that. I didn't attempt to ask for a simulation for SBM. Let me clarify my questions.
> >
> > Theoretically, could you *prove* that this algorithm can recover SBM in the setting that $p=0.01$ and $q=0.00001$ and $n$ goes to infinity under some unbalancedness. For example, when $s_1/s_2 \geq (p/q)^2$, where $s_1$ is the size of the largest cluster, and $s_2$ is the size of the second largest.  Even if you can not prove it now, do you believe this is correct? If you feel that my calculation is incorrect, please let me know. By contrast, I think the original spectral algorithm won't have an issue in this setting.
> >
> > Empirically, I was asking whether you have run tests on unbalanced real datasets and compared them with existing (even spectral) algorithms.

---

> > > ### Author Response · Authors · 2023-08-15
> > >
> > > Thank you for the clarification on your questions regarding the imbalanced clusters, and apologies for the previous misunderstanding. We cannot prove that our algorithm recovers the ground truth clusters in the SBM setting that you mention, using the techniques in our paper.
> > >
> > > We don't have a strong feeling either way about whether this could be proved using another technique, and admit that we are not familiar with the existing proof techniques for proving the correctness of spectral clustering for the SBM. It would be interesting to consider the proof of correctness for the SBM with our algorithm.
> > >
> > > On the empirical side, we emphasize that the additional experiments we added during the rebuttal phase do demonstrate that our algorithm is able to recover imbalanced clusters in practice, although you are correct that the real-world datasets we use are not imbalanced.

---

> > > > ### Comment · Reviewer_6nQB · 2023-08-15
> > > >
> > > > Thanks for the authors' response. Overall, I would like to keep my rating unchanged since my main concerns have not been fully addressed.

---

### Official Review · Reviewer_8rfq · 2023-07-07

**Soundness:** 3 good
**Presentation:** 3 good
**Contribution:** 3 good
**Rating:** 5
**Confidence:** 3

**Summary:**

The paper deals with the problem of spectral clustering in networks. The paper proposes a fast spectral clustering method by utilizing the power method and random projection. The paper also provides theoretical guarantee of cluster recovery.

**Strengths:**

The main strengths of the paper are -

(1) The paper provides a fast spectral clustering algorithm using the power method and random projections.

(2) The paper provides theoretical results on cluster recovery based on the proposed method.

(3) The paper provides a simulation study of the method and applies the proposed method to a few real-world networks.

(4) The paper is well-written.

**Weaknesses:**

The main weaknesses of the paper are -

(1) The paper extends the recent breakthrough on o(log(k))-embedding of vertices for cluster recovery. The use of the power method as a proxy for eigenvectors is not particularly novel.

(2) The stopping criteria in Algorithms 1 and 2 are in terms of order of log(n/k). The dependence of the results on the constants is not clear.

**Questions:**

(1) How does the proposed algorithm depend on the sparsity of the network?

(2) What should be the choice of the constants in Algorithms 1 and 2 in terms of O(log(n/k)) and O(log(k))?

---

> ### Author Rebuttal · Authors · 2023-08-09
>
> Many thanks for your time and thoughtful review. We respond to your specific questions below.
>
> **Question 1.**
> We address the effect of the network sparsity in our top-level comment.
>
> **Question 2.**
> This is a good question, and it is an example of a place where theory and practice diverge.
> In theory, the required constants are the ones from the analysis of Makarychev et al. [22] and as is common for a theoretical result, the constants in the big-O notation are not calculated explicitly (and are themselves based on previous theoretical results with non-explicit constants!).
>
> As we mention in Remark 3.3, in practice we find that setting $l \triangleq \left\lceil \log(k) \right\rceil$ and $t \triangleq \left\lceil 10 \cdot \log(n / k) \right\rceil$ works well.

---

> > ### Comment · Reviewer_8rfq · 2023-08-18
> >
> > Thanks for the rebuttal. It answers some of my questions. But, my concern about the sparsity conditions still remains as the concentration of the adjacency spectrum for SBM networks does not hold below O(nlog(n)) degree regime. So, I am not sure about the spectral method working for O(n) regime.
> >
> > So, I will keep my rating the same.
> >
> > Ref: Chen, Yuxin, Yuejie Chi, Jianqing Fan, and Cong Ma. "Spectral methods for data science: A statistical perspective." Foundations and Trends® in Machine Learning 14, no. 5 (2021): 566-806.

---

### Official Review · Reviewer_2unS · 2023-07-09

**Soundness:** 3 good
**Presentation:** 3 good
**Contribution:** 2 fair
**Rating:** 6
**Confidence:** 5

**Summary:**

The paper proposes a new spectral clustering algorithm that  instead of computing $k$-leading eigenvectors, the authors proposes to use power method to embed the nodes into a $\log(k)$ dimension space, which saves computational time when $k$ is large. Theoretical guarantees on misclassification rate with constant probability is provided under balanced cluster setting. Experiments on simulated network and $k$-NN graphs on real datasets are provided to show the effectiveness of the proposed algorithm.

**Strengths:**

- The idea on projecting the nodes to $\log(k)$ random space for spectral clustering is novel and interesting.
- The paper is well written and easy to follow


**Weaknesses:**

- The theoretical guarantee is limited on balanced cluster case and it is not a high probability bound
- The real data experiments are not convincible as
  - The authors manually construct $k$-NN graphs from some real data, instead of directly using existing real graph data.
  - The authors did not study how sparsity of the network may affect the results. For example, real networks are usually very sparse, will the accuracy be hurt more for the proposed methods under this case? For sparse networks, computing $k$-leading eigenvectors is not that expensive, and the advantage on the proposed method on computing speed may not be that significant, from Table 1, the computing speed up on most of the datasets are not that remarkable.
  - It is not clear for real dataset how balanced are the clusters.

**Questions:**

- Could the authors have a study on why for Letter dataset $k$ and $\log(k)$ EIGS takes so much more time than PM $k$/$\log(k)$? Did you also use power method to compute the leading $k$-eigenvectors? Since Letter dataset has fewer nodes than other datasets, but slightly larger $k$, it is very strange that the competing time difference is so much than other datasets.
- Why do you not use misclassification accuracy for the experiments since it is the main theoretical result you have?

**Limitations:**

I do not see a section stating or broader impact of the work. The authors discussed about the limitations in conclusion section.

---

> ### Author Rebuttal · Authors · 2023-08-09
>
> Thank you for your time and thoughtful comments.
> We respond to each of your questions below.
>
> **Further study of the Letter dataset.**
> For computing the eigenvectors, we use the Implicitly Restarted Arnoldi Method (IRAM) provided by the eigsh method in the Python scipy library, which itself uses the ARPACK Fortran library.
> It is not immediately clear why this performs so poorly on the Letter dataset, although we observe that the spectrum of the $k$ nearest neighbour graph has many eigenvalues very close to $0$.
> It may be that this causes the IRAM algorithm to converge slowly, while our algorithm requires only to approximate a random vector in the space spanned by the bottom $k$ eigenvectors and does not need to distinguish between eigenvectors with very similar eigenvalues.
>
> **Evaluating with misclassification accuracy.**
> While the misclassification accuracy is used in the theoretical guarantee of the algorithm, it is difficult to compute since it requires iterating over $k!$ possible permutations of the clusters.
> Instead, we evaluate with the Adjusted Rand Index (ARI) and Normalised Mutual Information (NMI) which are the standard metrics for evaluating clustering tasks.
>
> **Balanced assumption.**
> We address the balanced cluster assumption in our top-level comment.
>
> **Probability bound.**
> Regarding the probability bound, we report a probability of success of $0.8$ in the paper.
> Note that by running $O(\log(1 / \epsilon))$ copies of the algorithm, the success probability can be boosted to $1 - \epsilon$, for any $\epsilon$.
>
> **Evaluation on $k$-NN graphs.**
> In practice, the most common application of spectral clustering is on similarity graphs constructed from real data, such as $k$-NN graphs or kernel similarity graphs.
> As such, we feel that these experiments are a fair test of practical applications.
>
> **Sparsity.**
> We address the effect of the network sparsity in our top-level comment.

---

> > ### Comment · Reviewer_2unS · 2023-08-14
> >
> > Thanks for the rebuttal, I have read it and will keep my evaluation.

---

### Official Review · Reviewer_1Mto · 2023-07-26

**Soundness:** 3 good
**Presentation:** 3 good
**Contribution:** 3 good
**Rating:** 6
**Confidence:** 3

**Summary:**

The paper proposes a new fast spectral clustering algorithm. The main idea of the algorithm is to, rather than embed the vertices according to the k eigenvectors of the graph Laplacian, use the power method to efficiently approximate a random projection of the spectral embedding. Then, as in the classical spectral clustering algorithm, a k-means clustering algorithm is used to partition the vertices into k clusters.

The paper starts by discussing relevant preliminaries such as conductance, a higher order Cheeger inequality, the k-means objective and the power method. Next, the result is proven that if the k-th eigenvalue of the matrix M is close to 1 then the power method can be used to compute a random vector in the span of the first k eigenvectors of M. The authors then present their fast spectral clustering algorithm. They proceed to give a bound on the number of misclassified vertices by their proposed algorithm, and analyze the time complexity of the method. Finally, the authors sketch the proof of their main result, building on a recent result by Makarychev et al [22] which states that a random projection of data onto O(log(k)) dimensions preserves the k-means objective function for all partitions of the data with high probability.

In the experimental section, the proposed algorithm is evaluated on synthetic data under the stochastic block model as well as some medium-sized real world datasets. The algorithm is compared to the classical spectral clustering algorithm using k as well as log(k) eigenvectors as well as the power method using k eigenvectors. The authors observe that the proposed method is much faster than competing methods when increasing the number of clusters and number of vertices in the stochastic block model. The same observation is done on the real world datasets, while achieving comparable clustering accuracy.

**Strengths:**

The paper proposes a new algorithm for fast spectral clustering which is simple and easy to implement. The key technical contribution is a proof that they can efficiently compute an approximation of a random projection on the spectral embedding without computing the spectral embedding itself. The proposed method improves on the algorithm by Boutsidis et al [4] both in theory (as it has better time complexity, and an upper bound on the total misclassified vertices is given), as well as in practice on a number of datasets. The improved efficiency of the algorithm allows it to be applied for problems with a large amount of vertices and clusters.

**Weaknesses:**

The main weakness of the paper is the limited experimental evaluation. The paper only compares to the method by Boutsidis et al [4] as well as the classical spectral clustering algorithm with k or log(k) eigenvectors. A plethora of algorithms have been proposed to do fast approximate spectral clustering. The paper even mentions the algorithms by Choromanska et al [4] and Yan et al [34] but an experimental comparison to these methods is missing. While the proposed method always outperforms the other considered methods in terms of runtimes, it is in some cases significantly worse in terms of ARI and NMI score (in particular for the MNIST, Fashion and HAR datasets). It would be interesting to see if some of the other competing methods achieve a better trade-off here.

Moreover, one of the main selling points of the algorithm is its better time complexity and scalability compared to standard spectral clustering. However, the evaluation is done on relatively small datasets of up to 70000 vertices. The paper could be made stronger by also showing the results on some larger datasets where standard spectral clustering is not applicable any more.

Finally, the paper might also benefit from an evaluation in terms of the conductance score since it is the underlying objective used in spectral clustering.


====
Edit after Rebuttal:

In their rebuttal, the authors have provided some additional experiments and pointed out some additional results in the paper. I have therefore increased my score.

**Questions:**

How does the method perform against other fast spectral clustering techniques such as Choromanska et al [4] or Yan et al [34]?

**Limitations:**

The paper discusses the limitations of the proposed approach which is that it leads to a trade-off in terms of clustering accuracy compared to classical spectral clustering. There are no potential negative societal impacts of the work.

---

> ### Author Rebuttal · Authors · 2023-08-09
>
> Thank you for your thoughtful review. We respond to your questions regarding the experimental evaluation below.
>
> **Comparison with additional methods.**
> Table 1 in the attached PDF includes additional experimental results for the KASP algorithm proposed by Yan et al. [34].
> We find that on real-world data, the Nystrom method proposed by Choromanska et al. [4] produces poor results with an ARI consistently below $0.01$.
> We note that in the original paper [4], only small toy datasets are used to evaluate the algorithm and conclude that this method is not practical for real-world data.
> We will include these results and some more discussion in the final version of the paper.
>
> **Comparison on larger datasets.**
> We highlight the experiments on data drawn from the Stochastic Block Model (SBM).
> In particular, Figure 2(a) compares the running time of spectral clustering on massive graphs with a large number of clusters.
> We observe that on a graph with 400,000 vertices and 400 clusters, our newly proposed algorithm is able to recover the ground truth clusters in under 20 seconds, while classical spectral clustering cannot be applied in this instance.
>
> **Evaluation in terms of conductance.**
> We include an evaluation in terms of conductance in Table 1 of the attached PDF.

---

> > ### Comment · Reviewer_1Mto · 2023-08-15
> >
> > Thank you for your response.
> >
> > The main concern in my rebuttal was the limited comparisons to competing methods. The authors addressed this concern by adding a comparison to the KASP algorithm by Yan et al [34] where one can see that the proposed method is at least competitive in terms of NMI and ARI scores, while in some cases being significantly better in terms of runtime. Moreover, they discussed that the method by Choromanska et al [4] is typically not applicable for real-world data. The authors also added a comparison in terms of conductance score, as suggested in my review.
> >
> > Regarding the scalability to larger datasets, while the experiment on real-world datasets was done only on datasets of up to 70000 vertices, the authors pointed out that in the experiment in Figure 2a) the approach was indeed evaluated on data drawn from a stochastic block model of up to 400000 vertices. In fact, in my initial review I overlooked the fact that when increasing the number of clusters, the number of vertices was increased as well. That was my mistake, thank you for pointing it out in your rebuttal.
> >
> > Thank you again for your rebuttal and the additional experiments. I will take all these issues into consideration and I will increase my score accordingly.

---

> > > ### Author Response · Authors · 2023-08-16
> > >
> > > Thank you for carefully considering our response and raising your score. We hope that you can edit the original review to reflect this change.

---

### Author Rebuttal · Authors · 2023-08-09

We thank the reviewers for their time and thoughtful reviews.
Here we respond to questions which were shared by more than one reviewer, and we respond to other questions individually with reference to figures in the attached PDF.

**Balanced clusters assumption.**
The balanced cluster assumption is not integral to the proof of the theorem and without the assumption, we can obtain the same result under a slightly tighter assumption on the $k$-way conductance $\rho(k)$. Specifically, for a graph without balanced clusters, if $\rho(k) = O\left(\epsilon \cdot k^{-1} \cdot \log(n / \epsilon)^{-1} \right)$ then we obtain the same guarantee as stated in Theorem 3.1.
We will add some discussion to the next version of the paper.

In the attached PDF, we include some additional experimental results on an SBM with imbalanced clusters as requested by Reviewer 6nQB.
These results demonstrate that in practice, our algorithm is able to recover imbalanced clusters.

**Effect of network sparsity.**
The asymptotic running time of our algorithm for computing the vertex embedding is $\widetilde{O}\left(m \cdot \epsilon^{-3}\right)$, where $m$ is the number of *edges* in the graph.
Thus, the running time of our algorithm scales linearly with the sparsity of the graph.
Moreover, we highlight that in our experiments on real-world data, we use the $k$ nearest neighbour graph with $k = 10$, which results in a sparse graph with $m = \Theta(n)$.

---

### Decision · Program_Chairs · 2023-09-21

**Decision:**

Accept (poster)

**Comment:**

This paper proposes an efficient approach for spectral clustering by projecting the graph's vertices into O(log(k)) dimensions with the power method. The reviewers appreciate the proposed approach's technical contribution and the paper's reliable experimental results. Since all the reviewers are positive for the paper, I recommend accepting the paper.